# BAST: Bayesian Additive Regression Spanning Trees for Complex Constrained Domain

**Zhao Tang Luo**
Department of Statistics
Texas A&M University
ztluo@stat.tamu.edu

**Huiyan Sang**
Department of Statistics
Texas A&M University
huiyan@stat.tamu.edu

**Bani Mallick**
Department of Statistics
Texas A&M University
bmallick@stat.tamu.edu

## Abstract

Nonparametric regression on complex domains has been a challenging task as most existing methods, such as ensemble models based on binary decision trees, are not designed to account for intrinsic geometries and domain boundaries. This article proposes a Bayesian additive regression spanning trees (BAST) model for nonparametric regression on manifolds, with an emphasis on complex constrained domains or irregularly shaped spaces embedded in Euclidean spaces. Our model is built upon a random spanning tree manifold partition model as each weak learner, which is capable of capturing any irregularly shaped spatially contiguous partitions while respecting intrinsic geometries and domain boundary constraints. Utilizing many nice properties of spanning tree structures, we design an efficient Bayesian inference algorithm. Equipped with a soft prediction scheme, BAST is demonstrated to significantly outperform other competing methods in simulation experiments and in an application to the chlorophyll data in Aral Sea, due to its strong local adaptivity to different levels of smoothness.

## 1 Introduction

Over the past few decades, data collected from complex constrained domains have attracted much attention in machine learning and spatial statistics. Domains with non-trivial geometries, such as irregular boundaries, sharp concavities, and/or interior holes due to geographic constraints (e.g., lakes and coasts), impose challenges on statistical modeling, as the Euclidean assumption underpinning many traditional statistical and machine learning methods no longer holds for data with intrinsic geometries.

In this paper, we consider nonparametric regression problems with features lying on constrained domains or, more generally, compact Riemannian manifolds. To be more specific, we model the response variable $Y(\mathbf{s}) \in \mathbb{R}$ at a location $\mathbf{s}$ on a compact Riemannian manifold $\mathcal{M}$ as

$$Y(\mathbf{s}) = f(\mathbf{s}) + \epsilon(\mathbf{s}), \quad \epsilon(\mathbf{s}) \overset{\text{iid}}{\sim} \mathrm{N}(0, \sigma^2), \tag{1}$$

for some unknown function $f : \mathcal{M} \to \mathbb{R}$ and noise variance $\sigma^2$. In many applications, the true function $f(\cdot)$ may not be globally smooth but has discontinuities/abrupt changes across some narrow boundary regions in the domain. For example, housing prices can be substantially different in two neighboring communities, and ocean chlorophyll data that are separated by a narrow peninsula can exhibit distinct spatial patterns. It is of great need to develop new methodologies that can both respect intrinsic geometries of the domain and capture complicated local discontinuity patterns in the true function.

There is growing literature on nonparametric regression and smoothing for complex domains. Spline smoothing methods [33, 22, 41, 43, 37] have been developed for data on constrained domains,

but most of them focus on domains in $\mathbb{R}^2$. Sangalli et al. [36] generalized the spline methods to constrained regions in $\mathbb{R}^3$. For more general Riemannian manifolds, kernel based smoothing models, including kernel regressions [32, 19] and local regressions [1, 7, 12], were developed. Gaussian process (GP) regression is another popular tool for nonparametric regression problems, and many works focus on developing valid covariance kernels on spheres [see 20, 16, 15, among others]. More recently, a few practical GP models for constrained domains and Riemannian manifolds were studied in the literature [24, 31, 4, 13]. However, most of the aforementioned approaches assume globally smooth true functions and thus may not fully adapt to the ones with local discontinuities.

Ensemble tree models [5, 6] have been widely used in traditional nonparametric regression problems. One prominent example is the Bayesian additive regression trees (BART) model [10], due to its versatility and capability of producing uncertainty measures. However, to our knowledge, ensemble methods have not been used for nonparametric regression on complex constrained domains, and almost all ensemble tree methods rely on binary decision tree partition models as their ensemble members (weak learners). Nevertheless, binary trees may not be ideal to capture possibly highly irregular partitions on complex domains as they can only make splits parallel to Euclidean coordinate axes. For instance, in the U-shape domain shown in Figure 1(c), a complicated and over-clustered binary treed partition is needed to approximate a simple partition with three clusters (marked by different colors). Moreover, the rectangular partitions do not comply to irregular domain constraints, which may cause the so called "leakage" problems on complex domains. With a similar partitioning idea, Menafoglio et al. [28] proposed a Voronoi tessellation based model to account for domain constraints, but their method imposes convexity restrictions on partitions. Most recently, spanning treed partition models have been demonstrated as an effective tool for characterizing partitions with flexible shapes [23, 39, 26], but their focus is on traditional two-dimensional Euclidean spaces and linear regression settings.

Our contribution in this paper is to propose a novel Bayesian additive regression spanning trees (BAST) model for nonparametric regressions on complex constrained domains with efficient Bayesian inference algorithms. The backbone of BAST is a new random spanning tree (RST) manifold partition model, which replaces binary decision trees in each weak learner. RST is capable of capturing irregularly shaped partitions with a small number of spanning tree edge cuts while respecting intrinsic geometries and domain boundary constraints. Equipped with a *soft* prediction scheme, we show that BAST achieves a superior prediction performance over other competing methods on various tasks, thanks to its strong local adaptivity to different levels of smoothness.

The rest of the paper proceeds as follows. In Section 2, we present the RST partition models on manifolds and develop a new Bayesian nonparametric regression model with RST ensembles. Section 3 discusses algorithms for Bayesian inference. In Section 4, we illustrate the model performance by simulation experiments and a real chlorophyll data set in Aral Sea. Section 5 concludes the paper with some discussions. Additional details on Bayesian inference, hyperparameter selection, and sensitivity analysis are provided in Supplementary Materials. The code of BAST is available at `https://github.com/ztluostat/BAST`.

## 2 Bayesian Nonparametric Regressions with Additive Spanning Trees

### 2.1 A Spanning Treed Partition Model on Manifolds

Our novel nonparametric regression model is built upon an ensemble of partitions of observations. In this subsection, we introduce a stochastic partition model for data on a compact Riemannian manifold via random spanning trees (RSTs), which will serve as a building block to develop the sum-of-spanning-trees model in Section 2.2.

Let $\mathcal{M}$ be a $d$-dimensional compact Riemannian manifold that is known *a priori*, and $\mathcal{S} = \{\mathbf{s}_1, \ldots, \mathbf{s}_n\} \subseteq \mathcal{M}$ be a finite set of locations on $\mathcal{M}$ where the data are observed. We are interested in partitioning $\mathcal{S}$ into several disjoint subsets such that each subset consists of nearby locations where the data are relatively homogeneous that can be modeled separately. For spatial data, it is often desired to impose contiguity constraints on partitions. Below, we introduce the notion of spatially contiguous partitions on a manifold based on a spatial graph whose edges encode the relationship of spatial adjacency or neighborhood. Let $\mathcal{G} = (\mathcal{S}, \mathcal{E})$ be a connected undirected graph with vertex set $\mathcal{S}$ and edge set $\mathcal{E}$ that connects $\mathbf{s} \in \mathcal{S}$ to its "close neighbors". The construction of spatial graphs on a manifold will be discussed later in this subsection. We say $\pi(\mathcal{S}) = \{\mathcal{S}_1, \ldots, \mathcal{S}_k\}$, where $\mathcal{S}_j \subseteq \mathcal{S}$ for

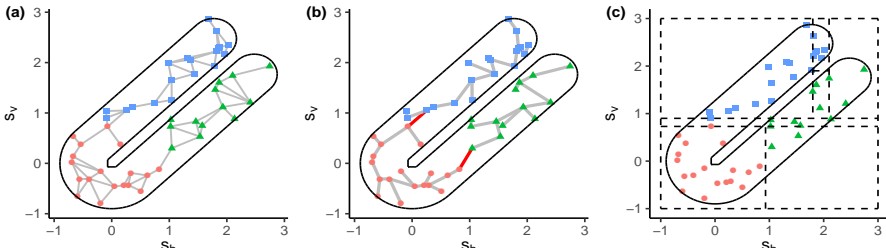

Figure 1: (a) A constrained Delaunay triangulation graph on a U-shaped domain. (b) A partition with three clusters obtained by removing the red edges in a spanning tree. (c) A binary treed partition nested in the three-cluster partition in (a, b).

$j = 1, \ldots, k$, is a *spatially contiguous partition* of $\mathcal{S}$ with respect to $\mathcal{G}$ if $\cup_{j=1}^{k} \mathcal{S}_j = \mathcal{S}$, $\mathcal{S}_j \cap \mathcal{S}_{j'} = \emptyset$ for all $j \neq j'$, and there exists a connected subgraph $\mathcal{G}_j = (\mathcal{S}_j, \mathcal{E}_j)$ of $\mathcal{G}$ for each $j$. We call each $\mathcal{S}_j$ a cluster, which consists of locations that are connected to each other and thus is spatially contiguous with respect to the spatial graph. Henceforth, when there is no risk of confusion, we will refer to spatially contiguous partitions simply as partitions. Figure 1(a) shows an example of a partition with three clusters.

For constrained domains in $\mathbb{R}^2$ such as the U-shaped domain in Figure 1, spatial graphs can be constructed via constrained Delaunay triangulations [CDTs; 8]. Specifically, let $\mathcal{G}_0$ be a CDT mesh on $\mathcal{S} \cup \mathcal{S}_B$, where $\mathcal{S}_B$ is a set of locations on the domain boundaries. Then the induced subgraph of $\mathcal{G}_0$ on $\mathcal{S}$ can be chosen as a spatial graph $\mathcal{G}$. Edges longer than a certain threshold can be removed if desired. See Figure 1(a) for an example of $\mathcal{G}$ constructed via CDT. For a general manifold, motivated by the nice adaptive properties of nonparametric regressions based on $K$ nearest neighbor ($K$-NN) graphs on manifolds [21, 27], one may construct $\mathcal{G}$ by a $K$-NN graph that connects $\mathbf{s} \in \mathcal{S}$ to its $K$ nearest neighbours with respect to geodesic distance.

Given $\mathcal{G}$, we model partitions on manifolds in a similar spirit as the spanning treed partition models developed for two-dimensional Euclidean spaces [23, 39, 26]. Specifically, a connected subgraph $\mathcal{T} = (\mathcal{S}, \mathcal{E}_\mathcal{T})$ of $\mathcal{G}$ is called a spanning tree of $\mathcal{G}$ if it has no cycle. A well-known property of spanning trees is that if a set of $k-1$ edges in $\mathcal{E}_\mathcal{T}$ is removed, we obtain a disconnected subgraph of $\mathcal{T}$ with $k$ connected components, which naturally defines a partition $\pi(\mathcal{S})$ with $k$ clusters by letting $\mathcal{S}_j$ be the vertex set of the $j$th component. In this case, we say $\pi(\mathcal{S})$ is *induced* by $\mathcal{T}$. See Figure 1(b) for an example. This property implies that we can simplify a complicated graph partition modeling problem to modeling spanning trees as well as the number and locations of removed edges.

Mathematically, conditional on $\mathcal{T}$ and $k$ we assume a discrete uniform distribution on all possible partitions induced by $\mathcal{T}$:

$$p\{\pi(\mathcal{S}) \mid k, \mathcal{T}\} \propto \mathbb{1}\{\pi(\mathcal{S}) \text{ is induced by } \mathcal{T} \text{ and has } k \text{ clusters}\}, \qquad (2)$$

where $\mathbb{1}(\cdot)$ is an indicator function.

Next, we consider a probabilistic model on the spanning tree space. Let $\omega_e$ be the weight for an edge $e \in \mathcal{E}$ and $\boldsymbol{\omega} = \{\omega_e : e \in \mathcal{E}\}$. We assume an iid uniform distribution on the edge weights and let $\mathcal{T}$ be the resulting minimum spanning tree (MST), i.e., the spanning tree with minimum $\sum_{e \in \mathcal{E}_\mathcal{T}} \omega_e$:

$$\mathcal{T} = \mathrm{MST}(\boldsymbol{\omega}), \quad \omega_e \overset{\mathrm{iid}}{\sim} \mathrm{Unif}\,(0, 1), \qquad (3)$$

where $\mathrm{MST}(\boldsymbol{\omega})$ denotes an MST of $\mathcal{G}$ based on the edge weights $\boldsymbol{\omega}$. Note that we are not assuming a discrete uniform distribution on the spanning tree space, with which it is challenging to sample spanning trees for Bayesian inference. As we will show in Section 3.1, our model specification leads to an exact and fast spanning tree sampler, taking advantage of the Prim's algorithm for MST constructions.

Finally, we assume a truncated Poisson distribution with mean parameter $\lambda_k$ on the number of clusters:

$$k \sim \mathrm{Poisson}(\lambda_k) \cdot \mathbb{1}(1 \leq k \leq \bar{k}), \qquad (4)$$

where $\bar{k}$ is the pre-specified maximum number of clusters.

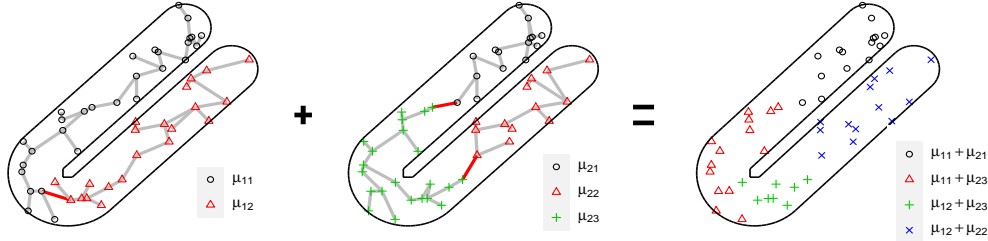

Figure 2: Demonstration of the partition obtained by adding two spanning treed partitions.

The following proposition states that the support of RST is rich enough to accommodate all possible spatially contiguous partitions on a manifold with no more than $\bar{k}$ clusters. The proof is postponed to Appendix C. Note that similar results do not hold for binary treed partition models. See Figure 1(c) for a counterexample where there does not exist a rectangle containing all the blue points without including any green or red ones.

**Proposition 1** *Let $\pi(\mathcal{S}) = \{\mathcal{S}_1, \ldots, \mathcal{S}_k\}$ be an arbitrary spatially contiguous partition. Then $\pi(\mathcal{S})$ is within the support of the partition model defined by* (2), (3), *and* (4) *if $k \leq \bar{k}$.*

### 2.2 A Sum-of-spanning-trees Regression Model

Given the data $\{Y(\mathbf{s}_i), \mathbf{s}_i\}_{i=1}^n$, we consider the nonparametric regression problem (1). Instead of assuming global continuity, we assume $f(\cdot)$ belongs to a broad class of piecewise smooth functions. We propose to model $f(\cdot)$ using a summation of weak learners based on the flexible RST partitions.

Given a partition $\pi(\mathcal{S})$ induced by $\mathcal{T}$ with $k$ clusters and cluster-wise constants $\boldsymbol{\mu} = (\mu_1, \ldots, \mu_k) \in \mathbb{R}^k$, we define a mapping from $\mathcal{S}$ to $\mathbb{R}$ as

$$g(\mathbf{s}|\pi, \mathcal{T}, k, \boldsymbol{\mu}) = \mu_j \quad \text{if } \mathbf{s} \in \mathcal{S}_j,$$

where we write $\pi = \pi(\mathcal{S})$ for conciseness. This piecewise constant function on $\mathcal{S}$ serves as a weak learner for $f(\cdot)$, which approximates $f(\cdot)$ by $\mu_j$ locally in $\mathcal{S}_j$. A Bayesian additive spanning trees (BAST) model is a summation of piecewise constant functions based on various spanning-treed partitions. Specifically, for a pre-specified $M \in \mathbb{N}$, BAST models $f(\cdot)$ as

$$f(\mathbf{s}) = \sum_{m=1}^M g(\mathbf{s}|\pi_m, \mathcal{T}_m, k_m, \boldsymbol{\mu}_m), \quad \mathbf{s} \in \mathcal{S}, \tag{5}$$

where $\pi_m = \pi_m(\mathcal{S}) = \{\mathcal{S}_1^m, \ldots, \mathcal{S}_{k_m}^m\}$ is a partition with $k_m$ clusters induced by a spanning tree $\mathcal{T}_m$ of $\mathcal{G}$ and $\boldsymbol{\mu}_m = (\mu_{m1}, \ldots, \mu_{mk_m})$. Although in principle the spatial graphs for each weak learners need not be identical, we focus on the case where they share a common $\mathcal{G}$ for simplicity.

For $\mathbf{s} \in \mathcal{S}$, the additive structure (5) implies that $f(\mathbf{s})$ equals the summation of the $\mu_{mj}$'s corresponding to the clusters from each weak learner that $\mathbf{s}$ lies in. Figure 2 illustrates a summation of two spanning treed partitions. This together with the shrinkage priors to be discussed in Section 2.3 allows each weak learner to explain a small amount of the variation in the response variable. The step function approximation also allows for capturing both smoothness and discontinuities/abrupt changes in $f(\cdot)$ [35]. In particular, our model is more efficient than some existing ensemble binary tree and smoothing methods in recovering irregularly shaped regions where discontinuities happen, thanks to the versatility of RST in capturing highly flexible cluster shapes for complex constrained domains.

### 2.3 Prior Regularization

Similar to BART [10], shrinkage priors play an important role in regularizing weak learners and preventing overfitting for BAST. In this subsection we discuss the specification of prior models, which admits the form

$$p\left(\{\pi_m, \mathcal{T}_m, k_m, \boldsymbol{\mu}_m\}_{m=1}^M, \sigma^2\right) = \left\{\prod_{m=1}^M p(\boldsymbol{\mu}_m|\pi_m, \mathcal{T}_m, k_m)p(\pi_m, \mathcal{T}_m, k_m)\right\} p(\sigma^2).$$

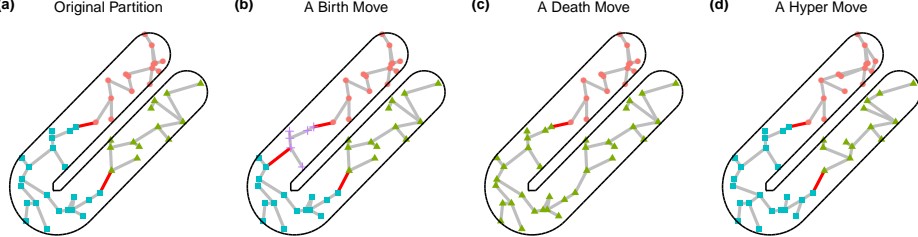

Figure 3: Partitions and spanning trees obtained after (b) a birth, (c) a death, or (d) a hyper move from the original partition and tree in (a).

We assign an iid RST prior for $(\pi_m, \mathcal{T}_m, k_m)$ given by (2), (3), and (4). Small values of $\lambda_k$ and $\bar{k}$ in (4) are typically chosen to restrict the number of clusters in each partition, leading to simpler piecewise constant structures in each weak learner which prevent overfitting and encourage better mixing in Markov chain Monte Carlo. Note that we regularize the number of clusters in a more direct way than binary treed partition priors [9, 10] which implicitly penalize large numbers of clusters by increasing the prior probability that a node is terminal as the depth of the node.

Rescaling $Y(\mathbf{s})$ into $[-0.5, 0.5]$, we opt to place a shrinkage prior that concentrates around 0 for $\boldsymbol{\mu}_m$ following [10]. Conditional on $(\pi_m, \mathcal{T}_m, k_m)$, we choose a conjugate prior for $\boldsymbol{\mu}_m$ given by

$$\boldsymbol{\mu}_m | \pi_m, \mathcal{T}_m, k_m \sim \mathrm{N}_{k_m}(\mathbf{0}, \sigma_\mu^2 \mathbf{I}_{k_m}), \tag{6}$$

independently for $m = 1, \ldots, M$, where $\sigma_\mu$ is assumed to depend on the number of weak learners, specifically, $\sigma_\mu = 0.5/(a\sqrt{M})$ with $a > 0$. Intuitively, when we have a larger number of weak learners, it is desired to impose stronger shrinkage effects by choosing a smaller $\sigma_\mu$ such that each learner is not too influential to the overall fit. A default choice of $a$ can be $a = 2$, which assigns $0.95$ prior probability for $f(\mathbf{s})$ that lies in $[-0.5, 0.5]$.

Finally, we assign a conjugate inverse-$\chi^2$ prior on $\sigma^2$: $\sigma^2 \sim \nu \lambda_s / \chi_\nu^2$. We fix $\nu = 3$ and choose $\lambda_s$ in a data-driven way such that the prior satisfies $\mathbb{P}(\sigma^2 < \hat{\sigma}^2) = 0.90$ similarly as in BART [10], where $\hat{\sigma}^2$ is the sample variance of $\mathbf{Y} = \{Y(\mathbf{s}_1), \ldots, Y(\mathbf{s}_n)\}$.

## 3 Bayesian Inference

### 3.1 Estimation

Inference of BAST is based on a tailored backfitting Markov chain Monte Carlo (MCMC) algorithm [18], in which we successively sample $(\pi_1, \mathcal{T}_1, k_1, \boldsymbol{\mu}_1), \ldots, (\pi_M, \mathcal{T}_M, k_M, \boldsymbol{\mu}_M)$, and $\sigma^2$ from their respective full conditionals. Our sampler for the full conditional $p(\pi_m, \mathcal{T}_m, k_m, \boldsymbol{\mu}_m | -)$, where $-$ stands for all other parameters and the data $\mathbf{Y}$, consists of two successive steps: (i) we first analytically integrate $\boldsymbol{\mu}_m$ out and sample from the collapsed conditional distribution for the partition parameters $p(\pi_m, \mathcal{T}_m, k_m | -)$, and (ii) we then sample $\boldsymbol{\mu}_m$ from $p(\boldsymbol{\mu}_m | \pi_m, \mathcal{T}_m, k_m, -)$. This design leads to better mixing and convergence performance of the sampler by avoiding the trans-dimensional problem for $\boldsymbol{\mu}_m$'s. Thanks to the conjugate priors, sampling from $p(\boldsymbol{\mu}_m | \pi_m, \mathcal{T}_m, k_m, -)$ and $p(\sigma^2 | -)$ follows standard procedures, and we leave the details to Appendix A.1.

Below, we focus on the sampling of the RST partition parameters. To draw samples of $(\pi_m, \mathcal{T}_m, k_m)$, one of the four moves — *birth*, *death*, *change*, and *hyper* — is performed with probabilities $r_b(k_m)$, $r_d(k_m)$, $r_c(k_m)$, and $r_h(k_m)$, respectively [26]. The first three moves modify the partition by proposing a new partition induced by the current spanning tree, and the hyper move updates $\mathcal{T}_m$ by sampling from its full conditional via an efficient sampling algorithm. Each move is detailed below and some of them are visualized in Figure 3.

In a birth move, one of the clusters is split into two by randomly removing an edge in $\mathcal{T}_m$ that connects vertices belonging to the same cluster. Denoting the new partition by $\pi_m^*$, the Metropolis-Hastings (M-H) acceptance ratio is given by

$$\min\left\{1, \frac{\lambda}{(k_m+1)} \times \frac{r_d(k_m+1)}{r_b(k_m)} \times \frac{\mathcal{L}(\mathbf{Y}|\pi_m^*, \mathcal{T}_m, k_m+1, -)}{\mathcal{L}(\mathbf{Y}|\pi_m, \mathcal{T}_m, k_m, -)}\right\}, \tag{7}$$

where $\mathcal{L}\left(\mathbf{Y}|\pi_m, \mathcal{T}_m, k_m, -\right)$ is the integrated likelihood with $\boldsymbol{\mu}_m$ marginalized out, whose closed form can be found in Appendix A.1. Opposite to the birth move, a death move randomly merges two adjacent clusters in $\pi_m$. Specifically, an edge in $\mathcal{T}_m$ that connects two distinct clusters in $\pi_m$ is uniformly selected and then the two clusters are merged into one. The M-H ratio is analogous to (7). In a change move, a death move is performed followed by a birth move, so that the number of clusters is unchanged. This move is designed to encourage better mixing of the sampler.

Finally, a hyper move updates $\mathcal{T}_m$ using an exact sampler, which adaptively learns a spanning treed spatial order so that we can obtain a partition that is more compatible to the homogeneity pattern of data in subsequent MCMC iterations. Specifically, we sample the edge weight $\omega_e$ of $\mathcal{G}$ from iid $\mathrm{Unif}(1/2, 1)$ if two endpoints of $e$ are in different clusters under $\pi_m$, and otherwise from iid $\mathrm{Unif}(0, 1/2)$. A new spanning tree is the MST generated by Prim's algorithm using the new edge weights. It can be shown that the resulting MST induces the current partition [39] and is an exact sample from its full conditional distribution [26].

The overall computational complexity per MCMC iteration is $O\big(M((1-r_h)n+r_h n \log n)\big)$, where $r_h$ is the probability that a hyper step is selected which takes $O(n \log n)$ using Prim's algorithm for CDT and $K$-NN graphs, and $O(n)$ is the computation complexity required when birth/death/change steps are selected because a closed form marginal likelihood without matrix inversion is available when calculating acceptance ratios. In practice, we suggest a small value of $r_h$ such as $0.1$ to reduce the computation and allow the algorithm to spend more iterations on learning a good partition compatible with the current tree. To further reduce computation complexity, we have done some preliminary exploration of using *different* but *fixed* spanning trees for each weak learner during MCMC (i.e., setting $r_h = 0$). As shown in Appendix B.1.3, this significantly speeds up the computation while the prediction performance remains comparable.

## 3.2 Prediction

The prediction at an unobserved location $\mathbf{u} \notin \mathcal{S}$ involves two steps. First, in each weak learner, we randomly assign $\mathbf{u}$ to one of its nearby clusters subject to the manifold constraints, using a *soft* prediction scheme in a similar spirit to Linero and Yang [25]. Second, the prediction is obtained by summing the constants corresponding to the clusters that $\mathbf{u}$ belongs to over all weak learners.

Specifically, to obtain cluster memberships for $\mathbf{u}$, we define its neighbor set $N_{\mathbf{u}} \subseteq \mathcal{S}$ as follows. For a constrained domain in $\mathbb{R}^2$, $N_{\mathbf{u}}$ is chosen as the vertices of the triangle containing $\mathbf{u}$ in the CDT mesh that belong to $\mathcal{S}$ (i.e., vertices on the domain boundary are excluded; see Appendix A.2 for detailed discussions). For general manifolds in higher dimensional spaces, $N_{\mathbf{u}}$ is specified as the set of $K$ nearest neighbors of $\mathbf{u}$ in $\mathcal{S}$ with respect to the geodesic distance.

Let $z_m(\mathbf{v}) \in \{1, \ldots, k_m\}$ be the cluster membership of a generic location $\mathbf{v} \in \mathcal{M}$ from the $m$th weak learner such that $z_m(\mathbf{s}) = j$ if $\mathbf{s} \in \mathcal{S}_j^m$. Intuitively, $\mathbf{u}$ is expected to share the same cluster membership as one of its neighbors in $\mathcal{S}$, and if $\mathbf{u}$ is near the boundary of a cluster in a partition, it is more ideal to assign $z(\mathbf{u})$ probabilistically to reflect the partitioning uncertainty and adapt for smoother functions. This motivates us to consider the following random assignment for $z_m(\mathbf{u})$'s: given $N_{\mathbf{u}}$ and a posterior sample of the partitions, $z_m(\mathbf{u})$ is sampled independently such that $\mathbb{P}\{z_m(\mathbf{u}) = z_m(N_{\mathbf{u},\ell})\} = \alpha_\ell$, for $\ell = 1, \ldots, |N_{\mathbf{u}}|$, where $N_{\mathbf{u},\ell}$ is the $\ell$th element in $N_{\mathbf{u}}$ and $\alpha_\ell$ satisfies $\sum_{\ell=1}^{|N_{\mathbf{u}}|} \alpha_\ell = 1$. One can specify $\alpha_\ell$ by setting $\alpha_\ell = 1/|N_{\mathbf{u}}|$ for all $\ell$, or via inverse geodesic distance weighting such as $\alpha_\ell \propto 1/d_g^b(\mathbf{u}, N_{\mathbf{u},\ell})$, where $d_g(\cdot, \cdot)$ is the geodesic distance and $b$ is some positive power.

At the second step, we sum $\mu_{m,z_m(\mathbf{u})}$ over $m = 1, \ldots, M$ to obtain a posterior predictive value of $\mathbb{E}\{Y(\mathbf{u})\}$ given samples of $z_m(\mathbf{u})$'s. A point predictor for $Y(\mathbf{u})$ can then be taken as the mean of the posterior draws, which allows us to average models with different RST partition structures.

Finally, we remark that the prediction algorithm is highly parallelizable, as the predictive sampling for each RST partition is independent.

Table 1: Prediction performance of BAST and its competing methods in the U-shape domain example. Standard errors are given in parentheses.

| | | BAST | BART | SFS | inGP |
|---|---|---|---|---|---|
| $\sigma = 0.1$ | MSPE | **0.189** (0.001) | 1.541 (0.075) | 0.418 (0.001) | 0.814 (0.002) |
| | MAPE | **0.188** (0.001) | 0.436 (0.010) | 0.340 (0.001) | 0.610 (0.001) |
| | Mean CRPS | **0.142** (0.001) | 0.380 (0.009) | — | — |
| $\sigma = 0.5$ | MSPE | **0.464** (0.006) | 1.704 (0.053) | 0.680 (0.007) | 1.057 (0.010) |
| | MAPE | **0.491** (0.004) | 0.686 (0.008) | 0.591 (0.004) | 0.752 (0.004) |
| | Mean CRPS | **0.371** (0.003) | 0.575 (0.007) | — | — |
| $\sigma = 1$ | MSPE | **1.283** (0.018) | 2.650 (0.056) | 1.491 (0.020) | 1.823 (0.025) |
| | MAPE | **0.888** (0.007) | 1.072 (0.008) | 0.951 (0.007) | 1.051 (0.008) |
| | Mean CRPS | **0.693** (0.006) | 0.889 (0.008) | — | — |

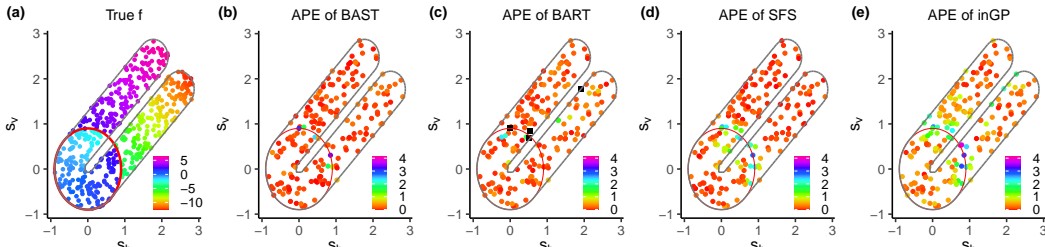

Figure 4: (a) True function in the training data. (b-e) APEs of one test data set with $\sigma = 0.1$. The discontinuity boundaries are marked as red circles. Black squares in (c) indicate APE > 4.10.

# 4 Experiments

## 4.1 U-shape Example

We first examine the BAST's performance of recovering piecewise smooth functions via some simulation experiments on a rotated U-shaped domain shown in Figure 4(a). Our true function $f(\cdot)$ is constructed based on the one in Ramsay [33], denoted as $f_R(\cdot)$. We create discontinuities along a circle of radius 0.9 centered at the origin as follows. For locations inside the circle, we flip $f_R$ by setting $f = -f_R$. For locations outside the circle, we set $f = 2f_R$ for those in the lower arm of the domain, and $f = f_R$ for those in the upper arm, such that the jump in the lower arm has a larger magnitude. We uniformly generate $n = 500$ locations in the domain as training data and 200 out-of-sample locations for prediction. Figure 4(a) shows the true function in the training data. The responses are generated according to (1) with different levels of noise $\sigma = 0.1, 0.5, 1$, and each noise level is replicated for 50 times.

The spatial graph $\mathcal{G}$ is constructed via CDT. We use $M = 20$ weak learners and set $\lambda_k = 4$ and $\bar{k} = 10$ to restrict the size of each partition. The prediction is based on the CDT graph, and we use inverse distance weighting with $b = 1$ to sample cluster memberships. The probabilities for MCMC moves are set as $r_b = r_d = r_c = 0.3$ and $r_h = 0.1$, with adjustments for cases where $k_m = 1$ or $\bar{k}$. We compare BAST with BART [10], and two other nonparametric regression methods for constrained domains, the soap film smoothing [SFS; 43] and the sparse intrinsic Gaussian process (inGP) regression [31]. For BART, we use the same number of weak learners as in BAST and use its default settings. We run the MCMC for both BAST and BART for 20,000 iterations, discarding the first half and retaining samples every 5 iterations. Our MCMC diagnostics suggest no convergence issue of BAST. We specify 32 equally spaced knots for SFS and set its basis dimension as 40. For sparse inGP, we use 24 equally spaced knots and simulate Brownian motions for 100,000 times. The prediction performance over 200 out-of-sample testing locations is assessed by mean squared prediction errors (MSPEs) and mean absolute prediction errors (MAPEs). For the two Bayesian approaches BAST and BART, we also compare their probabilistic prediction performance gauged by continuous ranked probability scores (CRPSs) based on their posterior predictive distributions [see, e.g., 14]. For all the metrics, lower values indicate better performance.

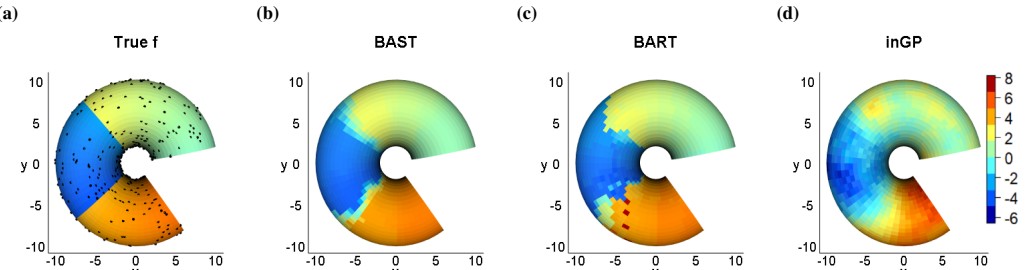

Figure 5: (a) True function and training locations (marked as black dots). (b-d) Predictive surfaces of BAST and its competing methods. All plots are viewed along the negative direction of the $z$-axis.

Table 1 summarizes the average performance metrics over 50 replicates for each noise level. BAST outperforms all its competitors in terms of all the metrics under all the noise levels. This is because BAST partitions the training data in a way that adapts to both the domain constraints and the irregularly shaped discontinuity boundaries, thanks to the flexible RST partitions. In contrast, the binary treed partitions adapt to neither types of boundaries, and neither of SPS and inGP captures discontinuities in the true function. This is also evidenced by Figure 4(b-e), where absolute prediction errors (APEs) of the test data in one replicate with $\sigma = 0.1$ are shown. The APEs from BAST are small for most locations except for those near the discontinuity boundaries. SFS also has similar patterns; however, errors near the discontinuity boundaries are much higher due to the global smoothness assumption in SFS. The general APE pattern for inGP is similar to SFS, except that it has larger errors, possibly due to the low-rank approximation of covariance functions. With the same number of weak learners, BART has larger errors near the upper discontinuity boundary, as more rectangular partitions are needed to well approximate irregular boundaries. Moreover, it also gives poor prediction at some locations near the domain boundary between the two arms, probably because binary treed partitions do not take into account the domain boundary when making axis parallel splits, and hence force some boundary locations in one arm to share the same cluster memberships with those locations in the other arm. We have also experimented using more weak learners in BART, and the results in Appendix B.1.1 suggest that BAST with a fewer number of weak learners still outperforms BART. Computation time for each method is reported and compared in Appendix B.1.3.

Hyperparameters of BAST can be tuned using standard cross-validation techniques. Our results in Appendix B.1.2 show that the fine-tuned BAST with respect to $M$, $\bar{k}$ and the shrinkage parameter $a$ for $\sigma_\mu$ achieves better performance than the default version in Table 1, but the performance of them is close to each other. We have also conducted additional sensitivity analyses to the hyperparameters $M$, $\bar{k}$, and $\lambda_k$ in Appendix B.1.2, which suggests that the performance of BAST is in general robust to them.

### 4.2   Bitten Torus Example

To illustrate BAST for more general manifolds, we consider a bitten torus example similar to Niu et al. [31]. A torus is a two-dimensional manifold embedded in $\mathbb{R}^3$ that is parameterized by $(\theta, \phi)$, where $\theta$ is the angle for the torus and $\phi$ is the angle for the tube. Let $R$ be the fixed distance from the center of the tube to the center of the torus, and $r$ be the fixed radius of the tube. The Cartesian coordinate $(x, y, z)$ on a torus can be written as $x = (R + r\cos\theta)\cos\phi$, $y = (R + r\cos\theta)\sin\phi$, and $z = r\sin\theta$. We create a bitten torus by setting $\phi \in [\pi/6, 1.7\pi]$ and $\theta \in [0, 2\pi]$.

We consider a piecewise smooth true function $f(x, y, z)$ defined on the bitten torus. We divide the torus into three subregions corresponding to $\theta \in [\pi/6, 3\pi/4]$, $\theta \in (3\pi/4, 5\pi/4]$, and $\theta \in (5\pi/4, 1.7\pi]$, respectively. The true functions in the first and the third regions are the same as the one used in Niu et al. [31], while we set the one in the second region as the negative of the function in Niu et al. [31], such that there are jumps along $\theta = 3\pi/4$ and $5\pi/4$. We generate responses at $n = 500$ random locations according to (1) with $\sigma = 0.1$ as training data. The true function and the training locations are shown in Figure 5.

We construct the spatial graph by a 10-NN graph based on the geodesic distance. Since the geodesic distance of a torus has no analytic form, we approximate it as in Isomap algorithm [40]. Specifically,

Table 2: Prediction performance of BAST and its competing methods in the bitten torus example. Standard errors are given in parentheses.

|  | BAST | BART | inGP |
|---|---|---|---|
| MSPE | **0.487** (0.002) | 1.115 (0.041) | 2.283 (0.005) |
| MAPE | **0.307** (0.001) | 0.406 (0.009) | 1.159 (0.003) |
| Mean CRPS | **0.225** (0.002) | 0.355 (0.008) | — |

we first construct a weighted, Euclidean distance based nearest neighbor graph on some fine grids and the training locations. Then we approximate the geodesic distance between two training locations by the length of the shortest path between them in the graph. For prediction at an unobserved location $\mathbf{u}$, we use its 5 nearest neighbors in $\mathcal{S}$ based on the geodesic distance as its neighbor set $N_{\mathbf{u}}$. We compare BAST with BART that uses Cartesian coordinates as features and sparse inGP that uses 24 equally spaced knots, as SFS is only applicable for domains in $\mathbb{R}^2$. Other model specifications are the same as those in Section 4.1.

As in the previous experiment, we compare the prediction performance at 200 random out-of-sample locations. The average performance metrics over 50 replicates in Table 2 suggest that BAST provides the most accurate prediction. We further compare the predictive surfaces of the three methods based on 2,500 grid points in Figure 5(b-d). As expected, both BAST and BART capture the piecewise structure in the true function, whereas inGP does not. In the interior of each subregion, BAST and BART approximate the true smooth functions fairly accurately. The major difference between the two methods occurs near the discontinuity boundaries. In the surface from BART, some partition boundaries are parallel to the Euclidean coordinate axes such as those near $x = -5$ and $y = \pm 2$, due to the use of binary trees, while this pattern does not appear in the one from BAST. Overall, the blue subregion recovered by BAST is more consistent with the truth. Notice that the estimated function at the discontinuity boundaries from BAST is smoother thanks to its soft prediction scheme. We have also experimented with different noise levels in Appendix B.2, and the findings are consistent.

## 4.3 Application to Chlorophyll Data

We apply BAST to analyze average remote sensed chlorophyll data in the Aral data over 1998-2002, which are available in the R package `gamair` [42]. The chlorophyll measurements at 485 equally spaced locations are shown in Figure 6(a). The southern part of the domain is separated by the isthmus of the peninsula near 59°E, and both shores of the peninsula have substantially different chlorophyll levels. It is thus desired to take into account this geographical constraints when modeling the data. The goal of our analysis is to assess how well BAST captures the patterns of chlorophyll and predicts for unobserved locations in this complex spatial domain.

We follow Niu et al. [31] to rescale the domain and model the chlorophyll level as a function of the scaled longitude and latitude plus some Gaussian noise. We use a same setup for BAST as in Section 4.1, except that we set $M = 30$ and $\bar{k} = 5$ to encourage smaller sizes of partitions as the number of weak learners is increased. For prediction, we sample the cluster membership of an out-of-sample location $\mathbf{u}$ using equal sampling probabilities $\alpha_\ell = 1/|N_{\mathbf{u}}|$. For BART, SFS, and sparse inGP, we also use the same settings as in the simulation experiments but with 30 trees for BART and 42 equally spaced knots for both SFS and inGP. The MCMC algorithms for BAST and BART are run for 30,000 iterations, keeping samples every 5 iterations from the second half.

We first compare the prediction performance of all the models via 10-fold cross-validation. In the end, the chlorophyll level at each observed location is predicted exactly once, and we compare MSPEs and MAPEs based on all locations in the data set. Similarly, the mean CRPSs over all locations are compared between BAST and BART. Table 3 shows prediction performance metrics for four models. BAST achieves the best performance among all the models.

Next, we turn to the predictive surfaces from each model, which are shown in Figure 6(b-e). All models capture the general patterns of the data. The predictive surfaces from SFS and inGP are fairly smooth, while some sharp jumps can be observed in the one from BART. The surface from BAST is somewhat in between, and preserves small-scale spatial dependence of the data. At the southern part of the eastern basin, BART identifies a rectangular region with high chlorophyll level, while the

Table 3: Prediction performance of BAST and its competing methods for the chlorophyll data.

|  | BAST | BART | SFS | inGP |
|---|---|---|---|---|
| MSPE | **2.346** | 2.933 | 2.894 | 3.191 |
| MAPE | **0.905** | 1.172 | 1.071 | 1.200 |
| Mean CRPS | **0.633** | 0.955 | — | — |

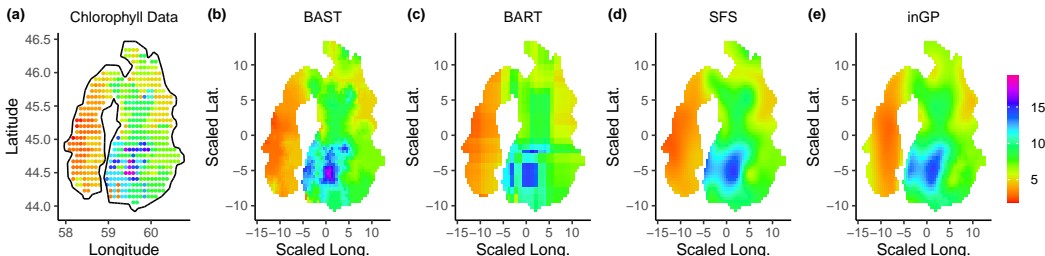

Figure 6: (a) Observed chlorophyll data. (b-e) Predictive surfaces from BAST and its competitors.

corresponding region obtained from BAST has irregular shape, which is more consistent with the data and the results from SFS and inGP. This is due to the highly flexible RST partition model that can give irregularly shaped clusters.

## 5 Conclusion and Discussion

In this paper, we developed a novel Bayesian nonparametric regression model on known manifolds and complex constrained domains using additive RST partitions. The RST weak learner enjoys flexibly shaped partitions while respecting the intrinsic geometries and domain constraints. The additive piecewise constant structure further allows BAST to approximate piecewise smooth functions with irregular boundaries of discontinuities, as evidenced by our simulation studies and real data analysis. In the case where the manifold is unknown, one may estimate the geodesic distance from the data [see, e.g., 29, and references therein] to construct spatial graphs. We leave this scenario for future research.

Similar to its binary treed counterpart BART, BAST is promising to serve as prior models in many other Bayesian hierarchical modeling settings, such as classification models with binary and multinomial responses [10, 30], survival analysis [2, 38], causal inference [17], and varying coefficient regressions [11]. As BAST is built upon a spatial graph, it is an interesting direction to extend our methodology for classification and regression on general graphs and networks [e.g., 3]. Finally, theoretical justifications are important but usually challenging for ensemble methods. For example, theoretical studies of BART have begun emerging only very recently [35, 34]. Posterior concentration results of BAST for estimating the true function can be potentially established in similar manners as BART, but non-trivial extensions are required to theoretically handle the complex spanning tree partition on manifolds and hence beyond the scope of this work.

This work has no foreseeable negative societal impacts, but users of BAST should be cautious of prediction outcomes based on ethically biased input data.

## Acknowledgments and Disclosure of Funding

The research was partially supported by NSF DMS-1854655, NSF CCF-1934904 and NIH R01AG064010. The authors thank the referees, the area chair, and Changwoo Lee from Texas A&M University for valuable comments.

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
