# Supplementary Materials of "BAST: Bayesian Additive Regression Spanning Trees for Complex Constrained Domain"

**Zhao Tang Luo**
Department of Statistics
Texas A&M University
ztluo@stat.tamu.edu

**Huiyan Sang**
Department of Statistics
Texas A&M University
huiyan@stat.tamu.edu

**Bani Mallick**
Department of Statistics
Texas A&M University
bmallick@stat.tamu.edu

These appendices provide supplementary details and results of BAST. Appendix A contains additional details on Bayesian estimation and prediction. Supplementary simulation details and results including hyperparameter tuning and computation time can be found in Appendix B. Finally, Appendix C provides the proof of Proposition 1.

## Appendix A    Details on Bayesian Inference

### Appendix A.1    Estimation

This appendix provides details on the Markov chain Monte Carlo (MCMC) algorithm discussed in Section 3.1. We use $\mathbf{g}_m$ to denote the $n$-dimensional vector of fitted values at the training locations $\mathcal{S}$ from the $m$th RST partition, that is, the $i$th element of $\mathbf{g}_m$ is $g(\mathbf{s}_i|\pi_m, \mathcal{T}_m, k_m, \boldsymbol{\mu}_m)$. Let $\mathbf{X}_{\pi_m}$ be an $n \times k_m$ binary matrix where the $(i, j)$th element is 1 if and only if $\mathbf{s}_i$ is in the $j$th cluster under the partition $\pi_m$. We write the partial residual term for the $m$th RST partition as

$$\mathbf{r}_m = \mathbf{Y} - \sum_{\ell \neq m} \mathbf{g}_\ell.$$

Recall that our MCMC algorithm proceeds by successively sampling $(\pi_1, \mathcal{T}_1, k_1, \boldsymbol{\mu}_1), \ldots,$ $(\pi_M, \mathcal{T}_M, k_M, \boldsymbol{\mu}_M)$, and $\sigma^2$ from their respective full conditional distributions. To sample from $p(\pi_m, \mathcal{T}_m, k_m, \boldsymbol{\mu}_m|-)$ for each $m = 1, \ldots, M$, we first sample the RST partition with $\boldsymbol{\mu}_m$ analytically integrated out, by performing a birth, a death, a change, or a hyper move with probability $r_b(k_m) = 0.3$, $r_d(k_m) = 0.3$, $r_c(k_m) = 0.3$, and $r_h(k_m) = 0.1$, respectively. Adjustments are made to the probabilities for the boundary cases where $k_m = 1$ and $k_m = \bar{k}$. This probability specification works well in our experiments, but one can modify it if desired. For the first three moves, the Metropolis-Hastings (M-H) acceptance ratio involves the integrated likelihood of $\mathbf{Y}$ given by

$$\mathcal{L}(\mathbf{Y}|\pi_m, \mathcal{T}_m, k_m, -) \propto |\mathbf{P}_{\pi_m}|^{-1/2} \exp\left(-\frac{1}{2}\mathbf{r}_m^\mathsf{T}\mathbf{P}_{\pi_m}^{-1}\mathbf{r}_m\right),$$

where $\mathbf{P}_{\pi_m} = \sigma^2\mathbf{I}_n + \sigma_\mu^2\mathbf{X}_{\pi_m}\mathbf{X}_{\pi_m}^\mathsf{T}$. The Sherman-Woodbury-Morrison formula is applied to simplify the computation of $\mathbf{P}_{\pi_m}^{-1}$ and $|\mathbf{P}_{\pi_m}|^{-1/2}$ as $\mathbf{X}_{\pi_m}\mathbf{X}_{\pi_m}^\mathsf{T}$ has a reduced rank $k_m$.

Conditional on a sample of $(\pi_m, \mathcal{T}_m, k_m)$, we sample $\boldsymbol{\mu}_m$ from $p(\boldsymbol{\mu}_m|\pi_m, \mathcal{T}_m, k_m, -)$, which is given by

$$[\boldsymbol{\mu}_m|\pi_m, \mathcal{T}_m, k_m, -] \sim \mathrm{N}_{k_m}\left(\mathbf{Q}_m\mathbf{b}_m, \mathbf{Q}_m\right),$$

where $\mathbf{Q}_m = \left(\frac{1}{\sigma^2}\mathbf{X}_{\pi_m}^\mathsf{T}\mathbf{X}_{\pi_m} + \frac{1}{\sigma_\mu^2}\mathbf{I}_{k_m}\right)^{-1}$ and $\mathbf{b}_m = \mathbf{X}_{\pi_m}^\mathsf{T}\mathbf{r}_m/\sigma^2$.

35th Conference on Neural Information Processing Systems (NeurIPS 2021).

Finally, we sample $\sigma^2$ from its inverse-gamma full conditional given by

$$[\sigma^2|-] \sim \text{IG}\left(\frac{n+\nu}{2}, \frac{1}{2}\left[\nu\lambda_s + \|\mathbf{Y} - \sum_{m=1}^{M}\mathbf{g}_m\|^2\right]\right),$$

where $\|\cdot\|$ is the Euclidean norm.

### Appendix A.2   Prediction in Two-dimensional Constrained Domains

In this subsection we provide details on specifying the neighbor set $N_{\mathbf{u}}$ for prediction at an unobserved location $\mathbf{u}$ in a constrained domain $\mathcal{M} \subset \mathbb{R}^2$. A constrained Delaunay triangulation (CDT) mesh can be constructed on $\mathcal{M}$ such that every unobserved location of interest is contained in a triangle. In the case where at least one triangle vertex is in $\mathcal{S}$, $N_{\mathbf{u}}$ is specified as those triangle vertices that belong to $\mathcal{S}$. Prediction at $\mathbf{u}$ is then performed as stated in Section 3.2.

In the extreme case where no triangle vertex is in $\mathcal{S}$, we choose $N_{\mathbf{u}}$ to be all the triangle vertices (which lie on the domain boundary). To sample the cluster membership of $\mathbf{u}$, we need to determine the cluster memberships for vertices on the domain boundary, which can be done by, for instance, assigning a boundary vertex to the same cluster as its nearest vertex in $\mathcal{S}$ with respect to the graph distance in the CDT mesh (when the number of vertices in the CDT graph is large, we expect this to well approximate the geodesic distance). Once we obtain the cluster memberships for boundary vertices, we can sample $z_m(\mathbf{u})$ from the cluster memberships of the vertices in $N_{\mathbf{u}}$ as in Section 3.2.

## Appendix B    Supplementary Simulation Results

We implement BAST in R and fit BART and SFS using R packages BART[1] [2] and mgcv[2] [3], respectively. The code for inGP is adopted from https://github.com/mu2013/Intrinsic-GP-on-complex-constrained-domain. Experiments are performed on a Linux machine with two Intel Xeon E5-2680 v4 processors and 64GB memory.

### Appendix B.1   U-shape Example

### Appendix B.1.1   Comparison to BART with Larger Numbers of Weak Learners

To demonstrate that BAST is more efficient than its binary treed competitors in recovering irregularly shaped regions where discontinuities happen in complex domains, we compare BAST with $M = 20$ to BART with various numbers of weak learners. The experiment setup is the same as in Section 4.1 except for the number of binary decision trees used in BART.

As shown in Table S1, BAST outperforms BART even when BART uses more weak learners, confirming that BART needs much more rectangular partitions to approximate irregularly shaped discontinuity boundaries, while BAST can recover them with only a few RST edge cuts.

### Appendix B.1.2   Hyperparameter Selection and Sensitivity

We consider selecting hyperparameters of BAST via cross-validation (CV) in the U-shape example with true noise standard deviation $\sigma = 0.1$. More specifically, for each replicate data set, we choose the number of weak learners $M$, the maximum number of clusters in each RST partition $\bar{k}$, and the shrinkage parameter $a$ that controls prior concentration around zero for $\boldsymbol{\mu}_m$ using 5-fold CV within the training data based on MSPE. The candidate values for each hyperparameter are summarized in Table S2, and a total of 18 hyperparameter combinations are considered for BAST. For comparision, we also choose the number of weak learners and the prior shrinkage parameter of $\boldsymbol{\mu}_m$ for BART using 5-fold CV, and their candidate values can be also found in Table S2.

Table S3 shows the performance of BAST and BART using the hyperparameters chosen by CV (referred to as BAST-cv and BART-cv, respectively). As a benchmark, the performance metrics for BAST and BART using the hyperparameters in Section 4.1 are also included (referred to as BAST-default and BART-default, respectively). The fine-tuned BAST-cv achieves better performance

---

[1]License: GPL (>= 2)
[2]License: GPL (>= 2)

Table S1: Prediction performance of BAST with $M = 20$ weak learners in the U-shape example. Results of BART with various larger numbers of weak learners $M$ are included for comparison. Standard errors are given in parentheses.

|  |  | BAST ($M = 20$) | BART ($M = 50$) | BART ($M = 100$) | BART ($M = 200$) |
|---|---|---|---|---|---|
| $\sigma = 0.1$ | MSPE | **0.189** (0.001) | 1.430 (0.049) | 1.302 (0.037) | 1.219 (0.036) |
|  | MAPE | **0.188** (0.001) | 0.408 (0.006) | 0.382 (0.005) | 0.380 (0.004) |
|  | Mean CRPS | **0.142** (0.001) | 0.353 (0.006) | 0.324 (0.004) | 0.318 (0.003) |
| $\sigma = 0.5$ | MSPE | **0.464** (0.006) | 1.694 (0.051) | 1.628 (0.039) | 1.532 (0.023) |
|  | MAPE | **0.491** (0.004) | 0.682 (0.007) | 0.695 (0.005) | 0.711 (0.005) |
|  | Mean CRPS | **0.371** (0.003) | 0.557 (0.006) | 0.553 (0.005) | 0.554 (0.004) |
| $\sigma = 1$ | MSPE | **1.283** (0.018) | 2.546 (0.054) | 2.441 (0.035) | 2.429 (0.032) |
|  | MAPE | **0.888** (0.007) | 1.085 (0.007) | 1.099 (0.007) | 1.120 (0.007) |
|  | Mean CRPS | **0.693** (0.006) | 0.870 (0.007) | 0.861 (0.006) | 0.870 (0.006) |

Table S2: Candidate values of hyperparameters for CV in the U-shape example.

| Method | Hyperparameter | Candidate values |
|---|---|---|
| BAST | # of weak learners $M$ | 20, 30, 50 |
|  | Maximum # of clusters per partition $\bar{k}$ | 5, 10 |
|  | $\mu$-prior shrinkage parameter $a$ | 1, 2, 3 |
| BART | # of weak learners $M$ | 50, 100, 200 |
|  | $\mu$-prior shrinkage parameter $a$ | 1, 2, 3 |

than BAST-default as expected, but the performance of them is close to each other, suggesting that BAST is robust to the choices of hyperparameters in this example. Both versions of BAST outperform BART with and without hyperparameter selection.

Next, we further investigate the sensitivity of the performance of BAST to hyperparameters $M$, $\bar{k}$, and $\lambda_k$ (the mean parameter of the truncated Poisson prior for $k$), and how they interact with each other. In general, for large $M$, one may prefer smaller $\lambda_k$ and $\bar{k}$ to prevent overfitting and encourage better mixing performance; for small $M$, one may afford larger $\lambda_k$ and $\bar{k}$ which may lead to better fitting. Below, we show additional simulation results with different values of $M$, $\lambda_k$, and $\bar{k}$ using the data set in Figure 4(b).

Table S4(a) shows the MSPE for various values of $M$ with a fixed $\lambda_k = 4$ and a fixed $\bar{k} = 10$. The prediction performance of BAST appears to be robust to $M$ except for extremely small $M$. Increasing $M$ slightly improves the performance until the training data is over-fitted. Next, we fix $\lambda_k = 4$ and examine the MSPEs for different combinations of $M$ and $\bar{k}$ shown in Table S4(b). Again, the performance of BAST does not appear to be sensitive to the choices of $M$ or $\bar{k}$. For a fixed $M$, increasing $\bar{k}$ improves out-of-sample performance until the model becomes too complex and overfits the training data. As expected, the optimal $\bar{k}$ for larger $M$ is smaller. Finally, we consider varying $\lambda_k$ while fixing $M = 20$ and $\bar{k} = 10$. As shown in the Table S4(c), the MSPEs for different values of $\lambda_k$ are comparable to each other, and the optimal MSPE is achieved with a moderate value $\lambda_k = 4$.

Table S3: Prediction performance of BAST and BART with and without CV in the U-shape example under noise level $\sigma = 0.1$. Standard errors are given in parentheses.

|  | BAST-cv | BAST-default | BART-cv | BART-default |
|---|---|---|---|---|
| MSPE | **0.186** (0.001) | 0.189 (0.001) | 1.277 (0.043) | 1.541 (0.075) |
| MAPE | **0.182** (0.001) | 0.188 (0.001) | 0.390 (0.005) | 0.436 (0.010) |
| Mean CRPS | **0.135** (0.002) | 0.142 (0.001) | 0.331 (0.005) | 0.380 (0.009) |

Table S4: MSPE of BAST under different settings of $M$, $\bar{k}$, and $\lambda_k$ in a U-shape domain data set with noise level $\sigma = 0.1$.

(a) MSPE under different values of $M$

| $M = 1$ | $M = 5$ | $M = 10$ | $M = 20$ | $M = 30$ | $M = 50$ |
|---|---|---|---|---|---|
| 25.54 | 0.203 | 0.196 | 0.192 | 0.186 | 0.188 |

(b) MSPE under different combinations of $M$ and $\bar{k}$

| | $\bar{k} = 5$ | $\bar{k} = 10$ | $\bar{k} = 15$ |
|---|---|---|---|
| $M = 20$ | 0.189 | 0.192 | 0.184 |
| $M = 30$ | 0.188 | 0.186 | 0.191 |
| $M = 50$ | 0.188 | 0.188 | 0.190 |

(c) MSPE under different values of $\lambda_k$

| $\lambda_k = 2$ | $\lambda_k = 4$ | $\lambda_k = 6$ | $\lambda_k = 8$ |
|---|---|---|---|
| 0.199 | 0.192 | 0.193 | 0.194 |

Table S5: Average computation time (in seconds) over 50 simulated data sets in the U-shape example under noise level $\sigma = 0.1$.

| BAST (in R) | BART | SFS | inGP |
|---|---|---|---|
| 651.49 sec. | 15.83 sec. | 0.68 sec. | 787.32 sec. |

### Appendix B.1.3    Computation Time

Finally, we report in Table S5 the average computation times (in seconds) of BAST and its competing methods over 50 simulated data sets in Section 4.1 with noise level $\sigma = 0.1$. The inference of BAST and BART is based on MCMC, and we remark that BART in the R package `bart` is implemented efficiently in C++ while BAST is implemented in pure R. The inference for SFS in the R package `mgcv` is based on an efficient optimization algorithm for point estimations only as opposed to a full MCMC inference with uncertainty quantifications, and hence achieves the fastest computation time. The model fitting of inGP requires expensive Brownian motion simulations and thus takes longer time than BAST does.

A more computationally efficient implementation of BAST is under active investigation. Our preliminary C++ implementation can reduce the computation time from 651.49 seconds to 53.58 seconds. As mentioned in Section 3.1, computation can be further improved by fixing spanning trees during MCMC. We refit BAST for the 50 simulated data sets in Section 4.1 with noise level $\sigma = 0.1$ by using *different* but *fixed* spanning trees for each weak learner. While the average prediction performance remains comparable (MSPE = 0.190, MAPE = 0.194, and mean CRPS = 0.145; also see Table 1 for baseline performance), the computation time is reduced to 16.82 seconds using the C++ implementation, which is comparable to BART.

### Appendix B.2    Bitten Torus Example

We consider the bitten torus example in Section 4.2 with two additional noise levels $\sigma = 0.5$ and $\sigma = 1$. The results are summarized in Table S6. Consistent to the findings under the noise level $\sigma = 0.1$, BAST performs the best among all three methods.

As in Appendix B.1, we also experiment with choosing hyperparameters via 5-fold CV for the data sets with true noise level $\sigma = 0.1$. In addition to the BAST hyperparameters in Table S2, we also select $K$, the size of the predictive neighbor set $N_{\mathbf{u}}$ discussed in Section 3.2, from its candidate values $\{3, 4, 5, 6\}$. As shown in Table S7, BAST outperforms BART in both CV and default settings. Our results again confirm that BAST performs reasonably well even without hyperparameter tuning.

Table S6: Prediction performance of BAST and its competing methods in the bitten torus example under different noise levels. Standard errors are given in parentheses.

|  |  | BAST | BART | inGP |
|---|---|---|---|---|
| | MSPE | **0.754** (0.008) | 1.358 (0.038) | 2.601 (0.033) |
| $\sigma = 0.5$ | MAPE | **0.584** (0.003) | 0.682 (0.006) | 1.240 (0.010) |
| | Mean CRPS | **0.405** (0.003) | 0.567 (0.006) | — |
| | MSPE | **1.568** (0.020) | 2.378 (0.050) | 4.628 (0.445)[*] |
| $\sigma = 1$ | MAPE | **0.960** (0.007) | 1.092 (0.009) | 1.648 (0.067)[*] |
| | Mean CRPS | **0.706** (0.006) | 0.904 (0.009) | — |

[*] The results for inGP under $\sigma = 1$ are based on 49 replicates due to numerical errors in one replicate data set.

Table S7: Prediction performance of BAST and BART with and without CV in the bitten torus example under noise level $\sigma = 0.1$. Standard errors are given in parentheses.

|  | BAST-cv | BAST-default | BART-cv | BART-default |
|---|---|---|---|---|
| MSPE | **0.463** (0.008) | 0.487 (0.002) | 0.850 (0.020) | 1.115 (0.041) |
| MAPE | **0.287** (0.004) | 0.307 (0.001) | 0.370 (0.004) | 0.406 (0.009) |
| Mean CRPS | **0.216** (0.003) | 0.225 (0.002) | 0.310 (0.004) | 0.355 (0.008) |

## Appendix C   Proof of Proposition 1

**Proof 1** *For any spatially continuous partition $\pi(\mathcal{S})$ with $k$ clusters, it follows from Propositions 2 of Luo et al. [1] that there exists a spanning tree $\mathcal{T}$ of $\mathcal{G}$ and a set of $k-1$ edges in $\mathcal{T}$ that induce $\pi(\mathcal{S})$. Hence, conditional on $\mathcal{T}$, the conditional probability for $\pi(\mathcal{S})$ is strictly positive due to (2) and (4). To show $\mathcal{T}$ is within the support of (3), note that $\mathcal{T}$ is the MST of $\mathcal{G}$ given the edge weights satisfying $\omega_e \in (0, 1/2)$ if $e \in \mathcal{E}_{\mathcal{T}}$ and $\omega_e \in (1/2, 1)$ if $e \notin \mathcal{E}_{\mathcal{T}}$. This completes the proof.*