# OpenReview forum: "BAST: Bayesian Additive Regression Spanning Trees for Complex Constrained Domain"
_NeurIPS.cc/2021/Conference — NeurIPS 2021 Poster_

### Official Review · Reviewer_R75X · 2021-07-06

**Rating:** 5
**Confidence:** 4

**Summary:**

Authors develop a novel Bayesian nonparametric regression model specially on known manifolds and complex constrained domains using additive RST partitions, which take advantage of flexibly shaped partitions. If the manifold is unknown, one should estimate it before applying the purposed method.

**Limitations And Societal Impact:**

There are a lot improvements can be made, a few are as followed:
- Authors should find a way to fix the issues mentioned in the main review.
- The number of tree should be larger, since it is essentially a truncation. Per the original paper, a recommended number is at least 50. However, the larger the better. Another way is to implement the Reversible-Jump MCMC.
- Authors should examine whether BAST and BART converge or not after the purposed 20,000 iterations.

**Main Review:**

- The main contribution of this paper is replacing the binary decision tree in BART model with random spanning tree, a flexibly shaped partitions. In this sense, the novelty is somehow limited per the standard of NeurIPS.

- The experiments in the paper are not especially well-designed. For example, standard error of BAST goes up dramatically as $\\sigma$ increases, while BART told a different story. Authors should look into this or at least give some explanations.

- The paper is well written with a clear mind. Readers outside this narrow area should find it readable.

- Since the authors do not compare their method against some BART variants adapting to graph problem, it is hard for me to evaluate the significance of their work. In general, methods taking advantage of extra information (structure) should outperform those does not.

**Time Spent Reviewing:**

4

---

> ### Author Response · Authors · 2021-08-10
> **Author Response to Reviewer R75X**
>
> We thank the reviewer for carefully reviewing our manuscript.
>
> 1. Clarification on contributions:
> To the best of our knowledge, BART models and its variants have not been used for non-parametric regression problems on complex constrained domains,  and the vast majority of additive models rely on binary decisions tree weak learners. Our work offers a novel attempt that builds additive models based on spanning tree partitions to relax the rectangular shaped partitions imposed by decision tree models. We are very encouraged by the substantial improvements in numerical studies over state-of-the-art competitors for manifold non-parametric regression models. Due to the flexibility of our Bayesian graph-based approach, we hope it can potentially offer a new research direction and motivate many possible extensions in the popular area of ensemble methods.
>
>     Our weak learner is extended from the very recently developed random spanning tree (RST) partition model by [Luo2021]. The original RST model is only developed for a linear regression setting on a simple 2-D Euclidean space rather than the manifolds we considered. Although both are called tree models, the methodology and computation of RST differs from binary decision tree based weak learners (Bayesian CART) in many aspects. In RST, each node of the spanning tree represents a point in the domain, and the merge/split of an edge represents fusion/separation of clusters, whereas the node of the decision tree represents an input of the function and a split corresponds to an axis parallel split. The number of clusters is a parameter in BAST for which we directly assign a truncated Poisson prior. In contrast, BART does not explicitly specify the number of clusters as a model parameter, but instead assuming that the prior probability of a node being terminal is increasing as the depth of node, which implicitly penalizes large numbers of clusters. We will revise the manuscript to clarify this part.
>
>
> 2. We agree with the reviewer that the standard errors of prediction performance metrics are usually expected to increase as the true noise level $\sigma$ increases, as observed in our BAST model and two other baselines. We thank the reviewer for pointing out a very interesting phenomenon that the standard errors for the BART baseline behave differently from the other three methods and remain comparable as $\sigma$ increases.
> After reexamining our numerical results, we notice that the prediction metrics of BART are dominated by some locations (mostly near cluster boundary and domain boundary) where the prediction from BART has fairly large errors in most of the replicate data sets (e.g., the black location in the lower arm of the U-shape domain in Figure 4(c)).
> The large errors of BART at these locations are mainly because BART ignores intrinsic geometry and only makes axis parallel splits that may cause misclustering.
> The variability of errors at these locations remains similar as $\sigma$ increases, as they are mostly determined by the difference between the responses at these locations and the ones at their nearby regions. On the other hand, this is less of an issue for BAST and other competing methods that respect the intrinsic geometry of the domain and produce smoother prediction. We will add the Github link to our data and code for reproducibility in the revised manuscript.
>
>
> 3. We have searched the literature surrounding BART, but were unable to locate references related to 'BART variants adapting to graph problems' that the reviewer mentioned. We will really appreciate and be more than willing to cite and compare with those methods if the reviewer may guide us to some specific references.
>
>
> 4.   The ideal choice of the number of trees depends on many factors such as smoothness of true functions, constraint on the number of clusters for each weak learner, and computational resources. Therefore, we agree with the reviewer that it is possible to treat the number of trees as an unknown parameter, but do not fully agree with the reviewer that a larger number of trees should always be better because it may cause overfitting for certain true functions.  In general, our method requires less number of trees than BART, due to its adaptivity to the domain constraints and the irregularly shaped discontinuity boundaries in each weak learner. For the U-shape example, the results in Appendix B.1 suggest that BAST with a fewer number of weak learners still outperforms BART. For this reason, the recommendation given by BART regarding the number of trees may not apply in our case. As the reviewer suggested, we can either use metrics such as information criteria or cross-validations to select it or assign a prior distribution to sample from its posterior distributions. However, this will usually comes with an increased computational cost.
> Besides Appendix B.1, we have added additional sensitivity analysis to examine the effect of prior on the number of clusters and the number of trees. Please see our response to Comment 1 of Reviewer Kv2o. We will leave a full investigation to future research to weigh the trade-off between performance gains and computational costs.
>
>
> 5. We thank the reviewer for the suggestion. For BAST and BART, we have performed convergence diagnostics based on draws of $\sigma^2$ after burn-in and thinning from one data set in the U-shape domain example with true noise level $\sigma = 0.1$ (which is the same data set used in Figure 4(b, c)). Specifically, we consider Geweke's convergence diagnostic [Geweke1992] and Heidelberger and Welch's convergence diagnostic [heidelberger1983] implemented in the `R` package `coda` [Plummer2006]. Both diagnostics provide no evidence of convergence issues for BAST ($p$-value = 0.107 for Geweke's diagnostics and $p$-value = 0.571 in the stationarity test in Heidelberger and Welch's diagnostic). However, the chain from BART fails Geweke's diagnostic ($p$-value = 0.017) and Heidelberger and Welch's diagnostic suggests that the chain from BART requires a longer burn-in period to achieve stationarity.
>
>     Based on the evidence, we have re-run the BART baseline for all the $50$ data sets in the U-shape domain example with true noise level $\sigma = 0.1$. The MCMC algorithm of BART was run for $40,000$ iterations, with the first half as the burn-in period, and we thinned the chain every $5$ iterations. The following table summarizes the performance. The performance of BART with $40,000$ iterations is comparable to the one with $20,000$ iterations, and BAST with $20,000$ iterations still considerably outperforms both versions of BART.
>
> |     | BAST (20000 iterations)| BART (40000 iterations)| BART (20000 iterations)|
> |:----|--------------------------:|--------------------------:|--------------------------:|
> |MSPE |               0.189 (0.009)|            1.550 (0.526)|              1.541 (0.530)|
> |MAPE |               0.188 (0.007)|            0.425 (0.064)|              0.436 (0.068)|
> |Mean CRPS |          0.142 (0.008)|            0.372 (0.060)|            0.380 (0.066)|
>
>
> [Geweke1992] Geweke, J. (1992). Evaluating the accuracy of sampling‐based approaches to the calculation of posterior moments. In Bayesian Statistics 4, Bernardo, J. M., Berger, J. O., Dawid, A. P. and Smith, A. F. M. (eds.), 169‐193. Oxford: Oxford University Press.
>
> [Heidelberger1983] Heidelberger, P., & Welch, P. D. (1983). Simulation run length control in the presence of an initial transient. Operations Research, 31(6), 1109-1144.
>
> [Luo2021] Luo, Z. T., Sang, H., & Mallick, B. K. (2021). A Bayesian Contiguous Partitioning Method for Learning Clustered Latent Variables. Journal of Machine Learning Research, 22, 37-1.
>
> [Plummer2006] Plummer, M., Best, N. , Cowles, K., & Vines, K. (2006). CODA: Convergence Diagnosis and Output Analysis for MCMC, R News, vol 6, 7-11

---

> > ### Author Response · Authors · 2021-09-01
> > **Further comments or feedback?**
> >
> > Dear reviewer,
> >
> >    We are very grateful to the reviewer for the valuable and constructive comments and suggestions, which greatly helped us to improve the manuscript. We have provided a point-by-point response to the comments. We are wondering if there are any other comments or feedback to help further improve our work. Your time is much appreciated.
> >
> > Best regards
> > Authors of this paper

---

### Official Review · Reviewer_Kv2o · 2021-07-06

**Rating:** 8
**Confidence:** 5

**Summary:**

This paper proposes a modification of Bayesian Additive Regression Trees tailored to complicated or constrained input space.
The basic idea, like BART, is to learn a collection of piecewise-constant weak learners to approximate an unknown regression function. Unlike BART, which parametrizes each weak learner using a binary regression trees with axis-aligned splits, the proposed BAST procedure parametrizes the weak learners using a spatially-contiguous partition of the input space that is index by a random spanning tree.
The proposed procedure works quite well on simulated and real world data. It represents a very interesting new direction in a growing literature around BART.

**Limitations And Societal Impact:**

Yes.

**Main Review:**

Originality: The individual components of BAST are not especially novel; there are very active communities studying both BART and its extensions and spatially contiguous partitions using spanning trees . The main novelty, then, of the paper is the effective combination of these two ideas.

Quality: On the whole, the ideas underlying BAST are sound and I commend the authors for a well-executed paper. A few minor questions about the method:

- Sensitivity to $\overline{k}$ & its interplay with $M$: Table S2 in the Supplementary Materials suggests that, at least for the two synthetic data experiments, BAST is not especially sensitive to the choice of the upper limit on number of clusters per partition. It would be helpful, however, to probe the sensitivity a bit more. E.g. how does the performance change as one increases or decreases $\overline{k}$? I suspect that if one were to increase $M$, one could get away with a smaller choice of $\overline{k}$ (and vice versa). A more comprehensive sensitivity analysis would be greatly appreciated.

- Timing comparisons: I suspect that both BART and BAST are faster than the two other competing methods (SFS and inGP). Is this the case?

- Complexity: What is the per-iteration time complexity? Will the worst-case complexity be O(M * n^2) as you may have to re-run Prim's Algorithm every time you propose a "hyper" move? How does this compare to the competing methods (BART, e.g., will have per-iteration time complexity O(M * n))

Clarity: The paper is well-written.

Significance: I think this paper represents a good step in pushing the boundaries of BART and moving away from strict axis-aligned splits. I anticipate it will be quite useful in spatial modeling.


**Time Spent Reviewing:**

2

---

> ### Author Response · Authors · 2021-08-10
> **Author Response to Reviewer Kv2o**
>
> We thank the reviewer for carefully reviewing our manuscript.
>
> 1. The model complexity of BAST is jointly determined by the number of weak learners $M$, the mean parameter $\lambda_k$ in the truncated Poisson prior for the number of clusters per weak learner $k$, and the upper bound $\bar{k}$ of $k$. In general, for large $M$, one may prefer smaller $\lambda_k$ and $\bar{k}$ to prevent overfitting and encourage better mixing performance; for small $M$, one may afford larger $\lambda_k$ and $\bar{k}$ which may lead to better fitting. We have added additional simulations with different values of $M$, $\lambda_k$, and $\bar{k}$ using one data set in the U-shape domain example with noise level $\sigma = 0.1$.
>
>     Table R1 shows the MSPE for various values of $M$ with a fixed $\lambda_k = 4$ and a fixed $\bar{k} = 10$. The prediction performance of BAST appears to be robust to $M$ except for extremely small $M$. Increasing $M$ slightly improves the performance until the training data is over-fitted.
>
>     Next, we fix $\lambda_k = 4$ and examine the MSPEs for different combinations of $M$ and $\bar{k}$ shown in Table R2. Again, the performance of BAST does not appear to be sensitive to the choices of $M$ and $\bar{k}$. For a fixed $M$, increasing $\bar{k}$ improves out-of-sample performance until the model becomes too complex and overfits the training data. As expected, the optimal $\bar{k}$ for larger $M$ is smaller.
>
>     Finally, we consider varying $\lambda_k$ while fixing $M = 20$ and $\bar{k} = 10$. As shown in Table R3, the MSPEs for different values of $\lambda_k$ are comparable to each other, and the optimal MSPE is achieved with a moderate value $\lambda_k = 4$.
>
>     As a concluding remark, the parameters of BAST can be tuned using standard techniques such as cross-validation, even though our results suggest that BAST is in general robust to the choice.
>
> Table R1: MSPE of BAST for various values of $M$.
>
> | $M = 1$| $M = 5$| $M = 10$| $M = 20$| $M = 30$| $M = 50$|
> |-----:|-----:|------:|------:|------:|------:|
> | 25.540| 0.203|  0.196|  0.192|  0.186|  0.188|
>
>
> Table R2: MSPE of BAST for various values of $M$ and $\bar{k}$.
>
> |       | $\bar{k} = 5$| $\bar{k} = 10$| $\bar{k} = 15$ |
> |:------|---------:|----------:|----------:|
> |$M = 20$ |     0.189|      0.192|      0.184|
> |$M = 30$ |     0.188|      0.186|      0.191|
> |$M = 50$ |     0.188|      0.188|      0.190|
>
>
> Table R3: MSPE of BAST for various values of $\lambda_k$.
>
> | $\lambda_k = 2$| $\lambda_k = 4$| $\lambda_k = 6$| $\lambda_k = 8$|
> |---------------:|---------------:|---------------:|---------------:|
> |           0.199|           0.192|           0.193|           0.194|
>
>
> 2. The average computation time (in seconds) over $50$ data sets in the U-shape domain example with noise level $\sigma = 0.1$ is summarized in Table R4. All timings are based on a machine with two Intel Xeon E5-2680 v4 processors and 64GB memory. The inference of BAST and BART is based on MCMC, and we remark that BART in the `R` package `bart` is implemented efficiently in `C++` while BAST is implemented in pure `R`. The inference for SFS in the `R` package `mgcv` is based on an efficient optimization algorithm for point estimations only as opposed to a full MCMC inference with uncertainty quantifications, and hence achieves the fastest computation time. The model fitting of inGP requires expensive Brownian motion simulation and thus takes longer time than BAST does. A more computationally efficient implementation of BAST is under investigation.
>
>     We will add the comparison of computation times and the Github link to our data and code for reproducibility in the revised manuscript.
>
> Table R4: Average computation time.
>
> |   BAST|  BART|  SFS|   inGP|
> |------:|-----:|----:|------:|
> | 651.49 sec.| 15.83 sec.| 0.68 sec.| 787.32 sec.|
>
>
> 3. The overall computational complexity per iteration is $O\big(M((1-r_{h})n+r_{h}n\log n)\big)$, where $r_h$ is the probability that a hyper step (updating spanning trees) is selected which takes $O(n\log n)$ using Prim's algorithm, and $O(n)$ is the computation complexity required when a birth/death/change step is selected because a closed form marginal likelihood without matrix inversion is available when calculating acceptance ratios. In practice, we suggest a small value of $r_h$ such as $0.05$ to reduce the computation and  allow the algorithm to spend more iterations on learning a good partition compatible with the current tree.

---

> > ### Comment · Reviewer_Kv2o · 2021-08-22
> > **Thanks!**
> >
> > Thank you for your detailed responses to my questions. I think it would be helpful to comment on the computational complexity in the revised manuscript.

---

> > > ### Author Response · Authors · 2021-08-23
> > > **Thanks**
> > >
> > > We thank the reviewer for this suggestion. We will add the discussion on the computational complexity of our Bayesian algorithm in the revised manuscript.

---

### Official Review · Reviewer_kisJ · 2021-07-16

**Rating:** 7
**Confidence:** 2

**Summary:**

This paper proposes a new Bayesian additive regression spanning trees (BAST) model for nonparametric regressions on manifolds. The model uses a new random spanning tree manifold partition model as weak learners, which allows it to capture complicated local discontinuity patterns in the true function while respecting intrinsic geometries of the domain. Building on nice properties of spanning tree structures, an efficient Bayesian inference algorithm is developed. Experiments on simulation data and the real chlorophyll data in Aral Sea show that BAST significantly outperforms other competing methods.

**Ethical Concerns:**

None.

**Limitations And Societal Impact:**

Yes.

**Main Review:**

The paper is very well written and a pleasure to read. The Bayesian model and inference procedure in Sections 2-3 are clearly explained and technically sound. I found it is a solid piece of work as a whole and the BAST model is interesting itself and could be used in many applications.

Comments:
Line 79: The word "constrains" should be spelled as "constraints".
Line 120: Is would be useful to explain a bit more about the binary treed partition models, e.g., does it also need to specify the number k of clusters.

**Time Spent Reviewing:**

4

---

> ### Author Response · Authors · 2021-08-10
> **Author Response to Reviewer kisJ**
>
> We thank the reviewer for the accurate and positive summary of our proposed method.
> We have corrected the typo you pointed out.
>
> Regarding the number of clusters in each weak learner, this is indeed one of the important model components that differ between BART and BAST.
> The number of clusters is a parameter in BAST for which we directly assign a truncated Poisson prior. In contrast, BART does not explicitly specify the number of clusters as a model parameter. Instead, BART assumes that the prior probability that a node is terminal is increasing as the depth of the node, which implicitly penalizes large numbers of clusters. We will revise the manuscript to clarify this part.

---

### Official Review · Reviewer_Tbpc · 2021-08-02

**Rating:** 7
**Confidence:** 4

**Summary:**

This paper introduces a novel Bayesian model for nonparametric regression in complex spaces such as manifolds.  Let $S$ be the set of points $s$ at which the outcome $Y(s)$ is observed.  The basic idea is to take a known (or estimated) graph $G$ over $S$, and consider the class of piecewise constant functions over spanning trees of $G$.  The mean of the outcome $Y(s)$ at $s$ is then modeled as a finite sum of such functions $g_m$ for $m=1,\ldots,M$.  An MH-within-Gibbs MCMC algorithm is proposed, in which the parameters of each $g_m$ function are updated in turn by making a move that preserves the full conditional distribution.  A well-chosen prior on trees facilitates exact sampling from the full conditional on spanning trees, using a previously known result from the literature. On simulated and real data, the method performs very favorably compared to competitors, including BART, SFS, and inGP.

**Ethical Concerns:**

No ethical concerns.

**Limitations And Societal Impact:**

Yes.

**Main Review:**

I found the proposed model and inference technique to be quite interesting and thought-provoking.  The model appears to be novel, and while the individual pieces of the MCMC algorithm are not novel, the paper does a nice job of integrating these previous ideas.  The paper is well-written and clear, both in terms of exposition and mathematical presentation.  In terms of significance, if the good empirical results presented here generalize to other data sets, the method could be quite useful in many practical applications.

MAIN COMMENTS
1. Identifiability is not discussed in the article. Depending on how the model is used, this could be problematic.  For instance, if inferences on the individual component functions is desired, then there is a label-switching problem.  Also, lack of identifiability could affect MCMC mixing performance.
2. Out-of-sample predictions (Sec 3.2): The rationale for the proposed method of making out-of-sample predictions was not clear to me.  It seems like a natural approach would be to consider a new model in which the new point $u$ is included in $S$, and look at the posterior predictive on $Y(u)$.  Does this coincide with the proposed approach?
3. In Section 4.1, over what domain are the MSPEs and MAPEs computed?  For instance, are you averaging the prediction error over the whole domain $M$?  Or is this the error on the training set?  Or something else?
4. It seems like the MCMC sampler might mix slowly since each component is updated individually, holding the others fixed.  This might make it difficult for the sampler to move around the parameter space.  I would recommend using standard diagnostic techniques to assess MCMC convergence.
5. Computation time: How long does the BAST sampler take to run, compared to BART and the other competitors?
6. On what types of problems does BAST not perform well?  I would imagine that high-dimensional spaces might be more difficult (for BAST compared to BART, for example), since the distances between points are less useful for defining local smoothness in a high-dimensional space.  Evaluating performance on a high-dimensional example would make the paper much more compelling.
7. What about combining BAST with other types of functions, e.g., regression trees with axis-aligned splits like in BART?  A hybrid BAST-BART model might be interesting.

MINOR COMMENTS
- The paper assumes the data live on a compact Riemannian manifold. Is the compactness assumption needed?
- Line 100: I think “cycle” is a more standard term than “circle”.
- $M$ is used to denote both the manifold and the number of components in the model.



**Time Spent Reviewing:**

3

---

> ### Author Response · Authors · 2021-08-10
> **Author Response to Reviewer Tbpc**
>
> We thank the reviewer for carefully reviewing our manuscript.
>
> 1.  The cluster labels for each weak learner at each iteration are uniquely assigned, but it is possible that labels of clusters may switch during the run of RJ-MCMC. Nevertheless, samplings of spanning trees and piece-wise constants ($\mu_j$'s)  for each weak learner only rely on partitions of locations and hence are not affected by label switching. As a result, the label switching for each weak learner would not affect the updates of other weak learners. Similarly, our soft prediction method only requires the estimates of piecewiseconstants but not the labels of clusters.
>
>
> 2.  The prediction for an unobserved location $u$ involves two steps. First, the cluster membership of $u$ in each weak learner is softly determined by its nearest neighbors (subject to the intrinsic geometry) in the training data. Second, the prediction is obtained by summing the constants corresponding to the clusters that $u$ belongs to over all weak learners.
>
>     We agree with the reviewer that the current prediction method could be further improved. An interesting and promising direction is to use a finite element type of projections [Lindgren2011, Ferraccioli2021] to extend the current BAST model built on finite partitions of observed locations to an additive piecewise constant functional model defined for any arbitrary given points on manifolds. In this way, predictions and estimations can be done in a coherent framework.
>
>
> 3. The MSPEs and MAPEs in our simulation experiments are computed over $200$ out-of-sample locations that are randomly drawn from the domain and not used for model training.
>
>
> 4. We thank the reviewer for the suggestion. For BAST, we have examined the traceplot of $\sigma^2$ after burn-in and thinning from one data set in the U-shape domain example with true noise level $\sigma = 0.1$ (which is the same data set used in Figure 4(b)). The traceplot suggests no visible convergence issues. We have also performed Geweke's convergence diagnostic [Geweke1992] and Heidelberger and Welch's convergence diagnostic [Heidelberger1983] to the same chain using the `R` package `coda` [Plummer2006]. Both diagnostics provide no evidence of convergence issues ($p$-value = 0.107 for Geweke's diagnostics and $p$-value = 0.571 in the stationarity test in Heidelberger and Welch's diagnostic).
>
>
> 5. The average computation time (in seconds) over $50$ data sets in the U-shape domain example with noise level $\sigma = 0.1$ is summarized in the following table. All timings are based on a machine with two Intel Xeon E5-2680 v4 processors and 64GB memory. The inference of BAST and BART is based on MCMC, and we remark that BART in the `R` package `bart` is implemented efficiently in `C++` while BAST is implemented in pure `R`. The inference for SFS in the `R` package `mgcv` is based on an efficient optimization algorithm for point estimations only as opposed to a full MCMC inference with uncertainty quantifications, and hence achieves the fastest computation time. The model fitting of inGP requires expensive Brownian motion simulations and thus takes longer time than BAST does. A more computationally efficient implementation of BAST is under investigation.
>
> |   BAST|  BART|  SFS|   inGP|
> |------:|-----:|----:|------:|
> | 651.49 sec.| 15.83 sec.| 0.68 sec.| 787.32 sec.|
>
>
> 6. In many real applications, complex domains such as domains with complex geometries in 2D, line networks, torus, or sphere often do not  have a high dimension of inputs. As the reviewer conjectured, non-parametric regression with high dimensional inputs will be challenging if we naively apply current BAST without modifications. The main challenge lies in the construction of graphs for high dimensional input locations due to difficulty in defining (usually distance-based) neighbors, especially when many irrelevant inputs are included in the model.
> To accommodate high dimensional data, current BAST needs to be further improved by combining with some manifold/graph/distance metric learning and variable selection methods. For example, we may assume a Mahalanobis distance function with unknown scaling/rotation parameters for high dimensional points and use distance metric learning [Suarez2021] or Bayesian methods [Payne2020] to estimate the distance function and construct graphs.
>
>
> 7. What the reviewer suggested is another direction we can pursue to extend BAST for high dimensional inputs. We can have a hybrid approach to combine the merits of BART and BAST. For example, in each weak learner, we can use random spanning treed partitions on the domain of those inputs with a known structure (e.g., spatial domains) to more efficiently model partitions, and use binary decision treed partitions for unstructured inputs to handle variable selection and unknown scaling of inputs.
>
>
> 8. In this manuscript, we only considered a compact Riemannian manifold. We plan to develop posterior concentration results of BAST for estimating true functions, though non-trivial theories are required to handle the spanning tree partition on manifolds. We anticipate that requiring the compactness of manifolds may significantly ease the proof.
>
>
> 9. We have changed 'circle' to 'cycle', and corrected the notation issue with $M$ in the revised manuscript.
>
>
> [Ferraccioli2021] Ferraccioli, F., Arnone, E., Finos, L., Ramsay, J. O., & Sangalli, L. M. (2021). Nonparametric density estimation over complicated domains. Journal of the Royal Statistical Society: Series B (Statistical Methodology), 83(2), 346-368.
>
> [Geweke1992] Geweke, J. (1992). Evaluating the accuracy of sampling‐based approaches to the calculation of posterior moments. In Bayesian Statistics 4, Bernardo, J. M., Berger, J. O., Dawid, A. P. and Smith, A. F. M. (eds.), 169‐193. Oxford: Oxford University Press.
>
> [Heidelberger1983] Heidelberger, P., & Welch, P. D. (1983). Simulation run length control in the presence of an initial transient. Operations Research, 31(6), 1109-1144.
>
> [Lindgren2011] Lindgren, F., Rue, H., & Lindström, J. (2011). An explicit link between Gaussian fields and Gaussian Markov random fields: the stochastic partial differential equation approach. Journal of the Royal Statistical Society: Series B (Statistical Methodology), 73(4), 423-498.
>
> [Payne2020] Payne, R. D., Guha, N., Ding, Y., & Mallick, B. K. (2020). A conditional density estimation partition model using logistic Gaussian processes. Biometrika, 107(1), 173-190.
>
> [Plummer2006] Plummer, M., Best, N. , Cowles, K., & Vines, K. (2006). CODA: Convergence Diagnosis and Output Analysis for MCMC, R News, vol 6, 7-11
>
> [Suarez2021] Suárez, J. L., García, S., & Herrera, F. (2021). A tutorial on distance metric learning: Mathematical foundations, algorithms, experimental analysis, prospects and challenges. Neurocomputing, 425, 300-322.

---

### Decision · Program_Chairs · 2021-09-27

**Decision:**

Accept (Poster)

**Comment:**

This article introduces a novel Bayesian method for nonparametric regression on manifolds, referred to as Bayesian Additive Regression Spanning Trees (BAST). The basic idea is to build a sparse graph $\mathcal{G}$ on the observed covariate data points, and use a BART-like ensemble of tree-based models on $\mathcal{G}$.  The graph $\mathcal{G}$ captures the lower-dimensional geometry of the manifold on which the covariates live, and the tree-based models yield piecewise-constant functions on partitions of the data. A Gibbs sampler algorithm is provided for posterior inference. In particular, a well-chosen prior on spanning trees facilitates exact sampling from the full conditional on trees for each tree-based model component in the ensemble, using a previously known result from the literature. Experiments are performed on simulated and real data.

Generally speaking, the reviewers and I found the paper to be interesting and novel.  The paper is well-written and the method appears to provide a valuable contribution to this area of research.  The method provides a compelling improvement in performance relative to competing methods.

In my view, the main limitations are:

1) The computation time is quite costly compared to competitors.  For instance, on the U-shape example, BAST takes 651.49 seconds while BART takes only 15.83 seconds (see authors' reply to Reviewer Tbpc).  The authors mention that a more computationally efficient implementation of BAST is under investigation.

2) Performance is likely to degrade significantly as the dimensionality of the data grows; see point 5 by Reviewer Tbpc and the authors' reply.

3) The method for out-of-sample predictions seems *ad hoc* and could potentially be improved.  The method does not provide a coherent (i.e., projective) model for all points in the space, since it is defined only on the observed points due to reliance on the graph $\mathcal{G}$.  Thus, the model does not lead to a natural technique for making out-of-sample predictions, which is presumably why an *ad hoc* method was used.

The only score below the acceptance threshold was from Reviewer R75X, with a score of 5.  Although R75X has not responded further, I found the authors' reply to satisfactorily address the main criticisms and questions.

---

> ### Public Comment · ~Huiyan_Sang1 · 2022-07-15
> **Extension of BAST to high dimensional case**
>
> We thank the Reviewers (especially Tbpc) and AC for the valuable and thought-provoking comments!
>
> To address the second and third limitations in AC's comments,  we have published a new paper in ICML:
>
> Luo, Z.T., Sang, H. & Mallick, B.. (2022). BAMDT: Bayesian Additive Semi-Multivariate Decision Trees for Nonparametric Regression. Proceedings of the 39th International Conference on Machine Learning (long oral), Proceedings of Machine Learning Research, 162:14509-14526 Available from https://proceedings.mlr.press/v162/luo22a.html.
>
> Abstract: Bayesian additive regression trees (BART; Chipman et al., 2010) have gained great popularity as a flexible nonparametric function estimation and modeling tool. Nearly all existing BART models rely on decision tree weak learners with axis-parallel univariate split rules to partition the Euclidean feature space into rectangular regions. In practice, however, many regression problems involve features with multivariate structures (e.g., spatial locations) possibly lying in a manifold, where rectangular partitions may fail to respect irregular intrinsic geometry and boundary constraints of the structured feature space. In this paper, we develop a new class of Bayesian additive multivariate decision tree models that combine univariate split rules for handling possibly high dimensional features without known multivariate structures and novel multivariate split rules for features with multivariate structures in each weak learner. The proposed multivariate split rules are built upon stochastic predictive spanning tree bipartition models on reference knots, which are capable of achieving highly flexible nonlinear decision boundaries on manifold feature spaces while enabling efficient dimension reduction computations. We demonstrate the superior performance of the proposed method using simulation data and a Sacramento housing price data set.